# Understanding ecohydrology and biodiversity in aquatic nature-based solutions in urban streams and ponds through an integrative multi-tracer approach

Maria Magdalena Warter[1], Dörthe Tetzlaff[1,2,3], Chris Soulsby[1,3,4], Tobias Goldhammer[1], Daniel Gebler[5], Kati Vierrikko[6] and Michael T. Monaghan[7,8]

[1] Leibniz Institute of Freshwater Ecology and Inland Fisheries, Department of Ecohydrology and Biogeochemistry, Berlin, Germany
[2] Department of Geography, Humboldt University of Berlin, Berlin, Germany
[3] Northern Rivers Institute, University of Aberdeen, St. Mary's Building, Kings College, Old Aberdeen, Scotland
[4] Chair of Water Resources Management and Modeling of Hydrosystems, Technical University Berlin, Berlin, Germany
[5] Department of Ecology and Environmental Protection, Poznan University of Life Sciences, Poland
[6] Finnish Environment Institute, Built Environment Solutions Unit, Helsinki, Finland
[7] Leibniz Institute of Freshwater Ecology and Inland Fisheries, Department of Evolutionary and Integrative Ecology, Berlin, Germany
[8] Institute of Biology, Freie Universität Berlin, Berlin, Germany

*Correspondence to*: Maria Magdalena Warter (maria.warter@igb-berlin.de)

**Abstract.** Rapid urbanization and climate change affect ecohydrology, biodiversity and water quality in urban freshwaters. Aquatic nature-based solutions (aquaNBS) are being widely implemented to address some of the ecological and hydrological challenges that threaten urban biodiversity and water security. However, there is still a lack of process-based evidence of
ecohydrological interactions in urban aquaNBS, and their relationship to water quality and quantity issues at the ecosystem level. Through a novel, integrative multi-tracer approach using stable water isotopes, hydrochemistry and environmental DNA we sought to disentangle the effects of urbanization and hydroclimate on ecohydrological dynamics in urban aquaNBS and understand ecohydrological functioning and future resilience of urban freshwaters. Stable isotopes and microbial data reflected a strong influence of urban water sources (i.e. treated effluent, urban surface runoff) across stream NBS. The results show
potential limitations of aquaNBS impacts on water quality and biodiversity in effluent-impacted streams, as microbial signatures are biased towards potentially pathogenic bacteria. Urban ponds appear more sensitive to hydroclimate perturbations, resulting in increased microbial turnover and lower microbial diversity than expected. Furthermore, assessment of macrophytes revealed low diversity and richness of aquatic plants in both urban streams and ponds, further challenging the effectiveness of NBS in contributing to aquatic diversity. This also demonstrates the need to adequately consider aquatic
organisms in planned restoration projects, particularly those implemented in urban ecosystems, in terms of habitat requirements. Our findings emphasize the utility of integrated tracer approaches to explore the interface between ecology and hydrology, and provide insights into the ecohydrologic functioning of aquaNBS and their potential limitations. We illustrate the benefit of coupling ecological and hydrological perspectives to support future NBS design and applications, that consider the interactions between water and the ecosystem more effectively.


# 1 INTRODUCTION

Urban freshwater systems face rising anthropic and environmental pressures - from high concentrations of nutrients and other pollutants, to severe hydromorphological alterations and declining streamflow, resulting in reduced biotic richness and habitat degradation (Oswald et al., 2023; Richardson and Soloviev, 2021). In addition, rapid urbanization has been a dominant paradigm and source of unintended consequences for water quality  Still, urban waterbodies have major potential in contributing to climate mitigation and adaptation (Bartrons et al., 2024; Hack and Schröter, 2020; van Rees et al., 2023). As a result, changing perceptions in how urban waterbodies are valued, has led to a surge in stream restoration measures and use nature-based solutions (NBS), as a way to address some of the ecological and societal challenges that threaten biodiversity and human well-being in urban environments (Everard and Moggridge, 2012; Fletcher et al., 2024; van Rees et al., 2023).  Aquatic nature-based solutions in particular, such as rainwater retention ponds, wetlands or streams, are used to increase water security and reduce other water-related risks, making them key tools for urban climate mitigation and adaptation (Pinho et al., 2023).

In urban areas, aquatic nature-based solutions are often implemented to mitigate flood risk and increase water retention, enhance groundwater recharge, alleviate the urban heat island effect and support green spaces, while also delivering major amenity and health benefits for residents (Dorst et al., 2019; Seddon et al., 2020). However, as the need for urban climate resilience continues to rise, so do concerns about the effectiveness and sustainability of NBS approaches to tackle challenges at a low cost, while delivering benefits for nature and society (Nelson et al., 2020; van Rees et al., 2023; dos Reis Oliveira et al., 2020; Seddon et al., 2020). The lack of conclusive evidence and understanding of the role of NBS in improving climate resilience and local biodiversity, and their interaction with biological and hydrological conditions within water bodies, hampers a sustainable implementation of NBS that seek to maximize the synergies while limiting trade-offs between multiple ecosystem services (Pauleit et al., 2017; Raymond et al., 2017).

Safeguarding stream biodiversity remains crucial to sustaining wider ecosystem functioning and building climate resilience under rapid change (Van der Cruysse et al., 2024). Constructed wetlands and floodplain restoration, stream reconnection, urban gardens and re-vegetation of streambanks, as well as bioswales or retention ponds, can help improve impaired aquatic ecosystems, targeting different ecosystem services and functions (Davis and Naumann, 2017; Dorst et al., 2019; Stefanakis, 2019). However, in urban systems, the motivation for the implementation of small-scale NBS is often guided by simple ecological principles, such as local restoration of terrestrial and aquatic vegetation, habitat improvement, or specific local water management targets, such as water quality and flood control, while also delivering societal benefits (Van der Cruysse et al., 2024; Hack and Schröter, 2020; Hale et al., 2023). Such small-scale measures run the risk of not considering the overarching landscape-scale stressors and complex technological realities that define urbanized systems (i.e. hydrology, water chemistry, land use) (dos Reis Oliveira et al., 2020). In severely disturbed stream ecosystems, hydromorphological changes and/or flow management do not automatically guarantee the return and establishment of diverse ecological communities, especially if basic microbial processes, habitat, nutrient-biota relationships and aquatic food webs are not considered (Hilderbrand et al., 2023; Leps et al., 2016).

Sustaining the benefits that come with biodiverse and resilient natural system requires healthy stream microbiomes, which include bacteria, archaea and microalgae, as these are essential to fundamental ecosystem processes (Sehnal et al., 2021). Similarly, the adequate development of primary producers, such as large aquatic plants (macrophytes), is crucial, as they are an important component of the trophic network. Even though macrophytes may be in competition with other autotrophs, especially phytoplankton, they also provide habitat, shelter, and food for other organisms,, thus significantly contributing to the overall biodiversity of aquatic ecosystems (Chambers and Maberly, 2024). The positive impact of macrophytes also extends onto microbial communities, as links have been found between specific microbiomes and different habitat zones(i.e. leaf, roots, sediments, surrounding water), which are directly or indirectly provided by submerged macrophytes (Zhu et al., 2021). Similarly, relationships between microbial communities and emergent macrophytes root systems have been observed (Huang et al., 2020). Nevertheless, in urban freshwater systems, the multitude of biotic and abiotic stressors have not only led to the propagation of diverse anthropic microbial communities (McLellan et al., 2015; Vignale et al., 2023; Warter et al., 2024), but also created conditions that inherently limit macrophyte development (Gebler and Szoszkiewicz, 2022).

Given that aquatic microbes and plants are intrinsically linked to physicochemical conditions, and in particular eutrophication, they are particularly useful indicators of urban aquatic ecosystem health and functioning. Previous studies have identified macrophyte communities as reliable long-term indicators of water quality changes, incorporating changes in nutrient levels over annual (Gebler et al., 2017) or multi-year timescales (Jeppesen et al., 2005). Given their small size and rapid developmental cycles, microbial communities would be expected to respond more quickly to environmental changes. However, recent studies have shown that microbial responses to anthropogenic pressures also emerge on a seasonal or longer-term scale, rather than as immediate or short-term responses (Lavoie et al., 2018; Smucker et al., 2022).

Still, evidence of increased biodiversity in NBS is still scarce. Using microbial communities, such as planktonic bacteria, benthic algae and diatoms, in conjunction with macrophytes as integrative tracers of stream restoration and freshwater diversity, can strengthen our understanding of nutrient management and restoration effectiveness. The rapid advancement of novel molecular and DNA-based approaches (i.e. environmental DNA, eDNA) now allow for targeted and efficient monitoring and evaluation of broader ecological patterns (Klaus et al., 2015; Smucker et al., 2022). A growing number of studies used DNA-based tracers to link microbial communities to surface and subsurface flow paths (Klaus et al., 2015; Pfister et al., 2017), nutrient levels (Mansfeldt et al., 2020; Smucker et al., 2022; Zeglin, 2015) or urbanization and climate effects (Numberger et al., 2022; Warter et al., 2024). The number of studies using high throughput sequencing of microbial communities is rapidly increasing, particularly in the urban context where higher taxonomic resolution is important to disentangle anthropogenic influences on stream aquatic diversity and the hydrological cycle (Urycki et al., 2022). Even more, in conjunction with other physical and chemical tracers, such as stable water isotopes or major ions, information on water sources and flow paths, as well as nutrient levels and pollution sources can provide a robust tracer-based assessment of ecological, hydrological and chemical conditions (Kuhlemann et al., 2022; Marx et al., 2021; Warter et al., 2024).

In this study, we present empirical evidence, including stable water isotopes, microbial and macrophyte richness and diversity, and hydrochemistry, from established aquaNBS in the city of Berlin, Germany. The aim of this study was to understand how urbanization, hydrology and physiochemistry relate to microbial patterns and aquatic diversity in urban restored streams and ponds, and evaluate the implications for future NBS approaches and restoration of urban freshwater systems. More specifically, our objectives were: i) to characterize the main water sources and pathways, microbial patterns and macrophyte diversity in urban aquaNBS, including species richness, alpha and beta diversity; ii) identify the environmental drivers of microbial patterns and macrophytes and ii) evaluate the implications of catchment scale stressors on aquaNBS efficiency. Our findings contribute to a broader understanding of the role of microbial processes and urban water sources in urban stream restoration and offer critical evidence to support future design and implementation of urban aquaNBS that equally support biodiversity, ecology and societal goals.

## 2 DATA AND METHODS

The city of Berlin is located in the dry NE of Germany and receives, on average, 580 mm of annual precipitation (1991-2020 mean) (DWD,2023). Characteristic of this drier region, the majority is lost through evapotranspiration (~56%) (Limberg, 2007), with the remaining water distributed as urban storm drainage (12%), and groundwater recharge (27%). Berlin's water supply is provided through bankside filtration, involving extraction of surface water and groundwater from underlying aquifers. The majority of water is abstracted from the Spree river, which receives water from several local tributaries that enter the Spree from the north. Groundwater heads are higher near the Spree river (< 4 meters below ground level), and generally deeper (>10 meters below ground level) on the plateaus to the north. Despite the low recharge and comparatively high evapotranspiration, Berlin has numerous wetlands and a large number of surface water bodies, including small to medium-size lakes and ponds (~700) , 2023) (Fig. 1c).

The central area of Berlin is highly urbanized (~60%), with 35% impervious surfaces (Fig. 1d). There are also large areas of contiguous urban green space (12.1%) and forest (17.7%). Water quality and quantity of the major and smaller tributary rivers are characterized by land use and urban storm water drainage, combined sewer overflows and water abstractions (Möller and Burgschweiger, 2008). Catchment characteristics are shown in Table S1 in the supplementary material

### 2.1 Site Description

To capture the diversity of urban freshwater systems, a range of stream and pond sites were selected (Fig. 1a). Our goal for stream sites was to choose a representative set of sequential sites downstream, experiencing varying flow conditions, urbanization effects and surrounding land use. Furthermore., access and logistics must allow for regular sampling. Three local tributaries of the Spree, namely the Erpe, Panke, and Wuhle, were selected. Daily discharge for streams and water levels for ponds were obtained from publicly available data (SenUVK, 2024). The different streams are characterized by different land uses, discharge and restoration measures. For pond sites regular access was an important criterion for selection, along with

similar size, setting (i.e. park) and permanence. The full comparison included 12 study sites: eight streams and four urban ponds. Sites were further divided into three subgroups differentiated between i) streams with a substantial inflow of effluent from nearby wastewater treatment plants (WWTP), and ii) streams which do not receive effluent. This grouping was done in order to separate the effects of water sources on the physical and biological functioning of stream ecosystems (Fig. 1b). Finally, the three sample subgroups were: i) restored streams with effluent impact, ii) restored streams without effluent impact and iii) restored ponds (Fig. 1b). In the following and throughout the manuscript, we only refer to them as streams with or without effluent impact or ponds, as all of them are restored.

### 2.2.1 Urban streams – with effluent impact

Erpe and Panke are both strongly impacted by treated WWTP effluents discharging from upstream treatment plants (Fig. 1a). Effluent contributions to discharge vary seasonally, ranging between 80% during normal flows and 100% during low flow periods (Kuhlemann et al., 2021; Marx et al., 2021). Along both streams, successive stream restoration has been completed within the last 10 years to improve water quality and habitat conditions (SenUVK, 2009, 2013a). These included the removal of weirs, regular mowing of in-stream and bank vegetation, establishment of still-water zones and a return to near natural channel structures (SenUVK, 2013b). Along the Erpe (total catchment area: 222 km$^2$), two stream sites (Erpe HG and Erpe) in the lower catchment were selected within a 1.2 km reach. The river has a peri-urban character, with surrounding land use including agriculture and extensive wetlands (Fig. 1). Both Erpe sites are surrounded by high-density natural forest and a comparatively low degree of sealed surface, as they are in a less densely populated part of the catchment. Mean discharge is ~ 0.8 m$^3$/s (2001-2024), which is primarily comprised of effluent discharge from the WWTP Muenchehofe (mean: 0.5 m$^3$/s).

The Panke catchment (total catchment area: 220 km$^2$) is characterized by higher urban density, but also large amounts of green space and forest, especially in the upper catchment. The three sites are located along a 6 km reach in the lower catchment. The mean discharge is ~ 0.9 m$^3$/s (2008-2024), again with substantial effluent discharge (mean: 0.8 m$^3$/s) from the WWTP Schoenerlinde (Fig. 1a). The most upstream site (Panke SP) features a meandering channel design and fish steps, as well as re-established riparian zones. At the second site (Panke OL), located within a highly urban part of the catchment, stream banks were re-vegetated and stillwater zones were implemented. The third site (Suedpanke) is a side channel of the main river, where

an entire stream section (~ 800m) was redesigned as part of an urban green-blue structure to improve channel flow and habitat, runoff retention and provide green space for residents , 2009).

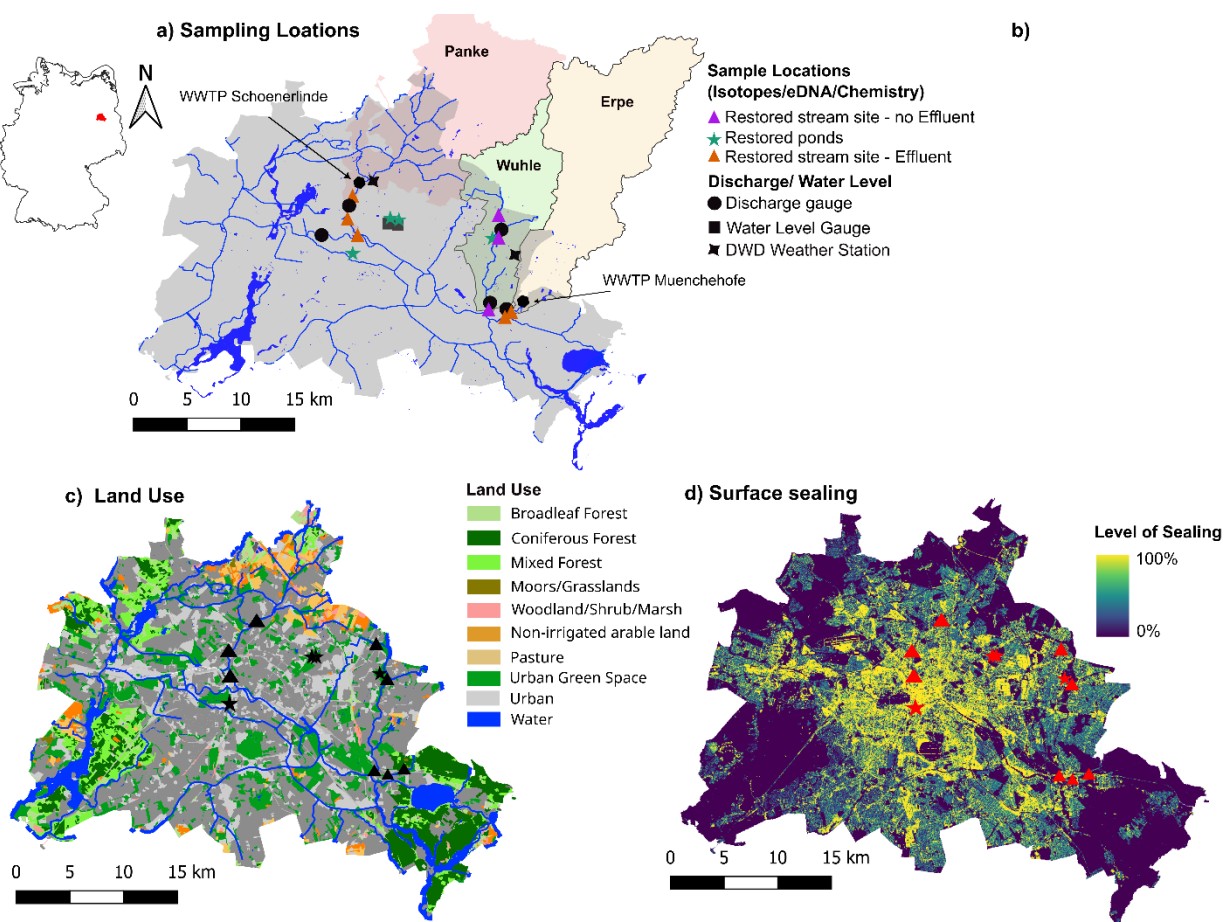

**Figure 1: a) Map showing Berlin sampling sites, where stable water isotopes, hydrochemistry and eDNA was sampled monthly between October 2022 - November2023. b) Urban freshwater sites were grouped into three subgroups: urban ponds, urban streams with effluent impact and urban streams without effluent impact. c) Map of Berlin showing land use and d) surface sealing levels, with sample sites indicated in black and red respectively. Data source for basemaps: Geoportal Berlin** (Geoportal Berlin/ALKIS, 2022)

### 2.2.2 Urban streams – without effluent impact

The river Wuhle (total catchment area: 100 km$^2$) primarily receives urban storm drainage and runoff (SenUVK, 2013b). Three sites were sampled along a 10km river reach, with the first two (Neue Wuhle and Wuhle) located in the upper catchment within a 4km and surrounded by extensive green space. The third (Wuhle OL) is located in the more urbanized lower catchment near the stream outlet. The stream does not receive treated effluent anymore (since 2003) (SenUVK, 2013b) and has been extensively restored. The course of the river and its streambed have been previously strongly modified through straightening,

deepening and the construction of weirs, to regulate flows. However, substantial efforts were undertaken to return the stream to more natural conditions and increase water retention and flood mitigation throughout the catchment. Mean discharge is ~0.3 $m^3/s$ (2002-2023), with sections of the river regularly drying out during the summer, especially in the upper catchment.

### 2.2.3 Urban ponds

Four representative urban ponds were included, each constructed or restored within the last 25 years. Pianosee (1.2 ha) is an
entirely artificial pond within a highly urbanized setting near Potsdamer Platz in central Berlin (~40% sealing, <10% green space). The pond has a maximum depth of ~1.8m (mean 0.5m) and is primarily used for rainwater retention, provision of greywater for nearby office buildings, and urban cooling. Excess rain water is stored in underground cisterns, with the majority of the water used to feed the pond. A two-stage mechanic-biologic filtration system cleans incoming rainwater and surface runoff. No water level data was available for this pond, as it is not regularly monitored. Obersee (3.7 ha) is an artificial pond
situated in a natural depression. It has a maximum depth of 3m (mean 1.49m) and serves primarily as a retention pond and to filter surface runoff. It is situated in an urban park (16% green space, 34% sealed), albeit in a highly urbanized area. The pond experiences frequent water quality issues, such as phosphorus overload and excessive silt accumulation. The shore is largely lined by concrete walls, along which reed belts, which were constructed to support natural filtration. Orankesee (4.1 ha) is in close proximity to Obersee (distance <100m), although the lakes are not connected. It is a natural lake fed by groundwater,
with a maximum water depth of ~6.5m (mean 2.6m). Half of the pond is closed off for swimming and reed belts and stillwater zones serve as habitat for water fowl. Water quality can be highly variable as the pond also receives urban runoff, but is generally better than in Obersee, due to the constant supply of freshwater. Finally, Wuhleteich (3 ha) is also an artificial pond located within a nature reserve (37% green space, 11% sealing) with a maximum water depth of ~ 2m (mean 0.75m). The pond is fed by precipitation and water diverted from the nearby Wuhle. The shore is vegetated by dense reed belts and hedges, and
pond substrate is very silty.

### 2.3 Stable water isotopes and hydrochemistry

Stable water isotopes, physicochemical parameters (pH, electrical conductivity (EC), dissolved oxygen (DO), water temperature) and solute tracers (i.e. major ions and anions) were sampled monthly from October 2022 to November 2023. For stable isotopes, water was filtered into 1.5 ml vials (LLG Labware) using 0.22 µm cellulose acetate filters. For major ions and
anions, grab water samples were also filtered (0.45µm cellulose acetate filter) before analysis. Stable water isotopes were analysed using Cavity Ring-Down Spectroscopy with a Picarro L2130i Isotopic Water Analyser (Picarro Inc., Santa Clara, CA). Physicochemical parameters were measured and recorded in the field using a handheld multiprobe (Multi 3630 IDS, WTW; Weilheim, Germany). Major ions and anions were used to characterize water source endmembers and flow paths. EC in particular was used as a proxy for the level of dissolved charged chemicals, to determine specific water or pollution sources.
Dissolved inorganic carbon (DIC) – as the sum of inorganic carbon species in water, was used to characterize water quality, ecosystem health and carbon cycling. Acid neutralizing capacity (ANC) was also calculated using a mass balance approach of

major anions and cations (Neal et al., 1999). Further details regarding sample preparation and analysis can be found in the Supplementary Material.

In addition, young water fractions (YWF) were calculated using open access code by Von Freyberg et al. (2018) after Kirchner (2016), using an iteratively re-weighted least squares (IRLS) fitted sine-wave approach. We used observed precipitation isotopes from Berlin Steglitz and stream and pond water isotopes to estimate the fraction of event water that reached the stream or pond within the previous 2-3 months. We compared sine-wave fit amplitudes of monthly isotope samples with amount weighted precipitation $\delta^{18}O$ and $\delta^2H$. To further assess evaporation effects on water isotopic conditions in streams and ponds, we used the line-conditioned excess (lc-excess), which indicates the offset of surface water samples from the Local Meteoric Water Line (LMWL) (Landwehr and Coplen, 2006). We use lc-excess as an environmental variable, in addition to water quality and hydrochemistry data, to evaluate the effect of evaporation and seasonal water loss on microbial community structures.

## 2.4 Microbial eDNA sequencing and analysis

Environmental DNA was collected seasonally in October 2022, January 2023, May 2024, and July 2024. Water samples (150 – 400 ml) were field-filtered through an Isopore polycarbonate membrane filter (Merck, Darmstadt, ø 47mm, pore size 0.22µm) using a hand-operated Nalgene vacuum pump and filter holder with 500-ml receiver (both Thermo-Fisher). Filters were preserved in 2-ml Eppendorf tubes filled with 96% ethanol and stored at -20°C until analysis. DNA extraction and community analysis of bacteria and diatoms/algae was carried out on an automated workstation biomek i7 hybrid (Beckman Coulter GmbH, Krefeld, Germany) at the Berlin Center for Genomics in Biodiversity Research as described by Warter et al (2024). Briefly, DNA was extracted using a DNeasy PowerSoil Pro Kit (Qiagen, Hilden, Germany) following the manufacturer's protocol (Handbook 03/2021, HB-2495-005) with the following exceptions: In step 1, 750 µl of CD1 buffer was used, and in step 16, CD6 buffer was replaced with AE buffer from the DNeasy Blood & Tissue kit (Qiagen). AE contains EDTA in order to provide a better conservation of the samples.

Sequences were analysed using cutadapt v 4.4 (Martin, 2013) and dada2 v 1.21.1 (Callahan et al., 2016) for R (v 4.2.0) (R Core Team, 2021). The resulting amplicon sequence variants (ASVs) were taxonomically identified using the PR2 v. 5.0.0 database (Vaulot et al. 2023) or RDP trainset 18 database (Callahan 2020). Bacteria were identified to genus level and diatoms/algae to species level. Unclassified ASVs ("NA" for species or genus level) and rare ASVs (occurring in < 100 total reads or present in fewer than 10% of the samples) were excluded from analysis to reduce influence of sequencing artefacts.

## 2.5 Macrophyte sampling

An aquatic vegetation survey was carried out at each site in September 2023. Each site included two macrophyte survey sites: i) a more natural and vegetated site, characterized by well-developed, undisturbed vegetation along the bank, and ii) a more artificial and less vegetated site, indicating more anthropogenic disturbances. Sites were selected using a stratified random sampling approach, considering the level of naturalness and alteration. At each site, the most and least altered quadrats, spanning a 10 by 10-metre square were selected. Quadrats were then divided into a terrestrial and aquatic zone, where i) an inventory of all species present was carried out (qualitative inventory) and ii) species abundance was estimated (quantitative inventory). For detailed sampling design see also Szoszkiewicz et al. ( 2025). The survey included all groups of vascular aquatic plants, including emergent, submerged, free-floating and floating rooted macrophytes identified at the species level. The abundance of each taxon was estimated using a 9-point level cover scale (e.g. Szoszkiewicz et al., 2020). Based on field data, species richness (N), relative abundance, and Shannon diversity were calculated for each site.

## 2.6 Statistical Analysis

All statistical analyses were performed using R Studio (version 4.2.3) (R Core Team, 2021). The vegan package was used to assess ecological data (Oksanen et al., 2022). Spatial and seasonal differences in physiochemical parameters and hydrochemistry were identified using Kruskal-Wallis testing ($\alpha = 0.05$). Assessments of microbial community patterns were performed using relative abundance data. Standard methods for the representation of multi-dimensional data were used, such as non-metric multidimensional scaling (NMDS), to evaluate spatial patterns in microbial diversity, using the Bray-Curtis dissimilarity index (Dexter et al., 2018). Community structures were assessed through pair-wise comparison analysis of similarities (ANOSIM). To characterize microbial $\alpha$-diversity, genus and species richness (i.e. total number of species present) was calculated for bacteria and diatoms/algae, respectively. We used the Shannon diversity (Shannon H Index), which takes into account the relative abundance of different species rather than just species richness, to evaluate the overall diversity of species in each subgroup (Hill et al., 2003). Differences in alpha diversity for microbial and macrophyte communities between subgroups were evaluated through nonparametric Kruskal Wallis tests (p-value of <0.05) and mixed two-way analysis of variance (ANOVA) (Wang et al., 2016).

Taxonomic $\beta$-diversity was assessed using the betapart package in R (Baselga and Orme, 2012), using relative abundance data transformed into binary presence/absence data (presence = 1, absence = 0). The Sorensen dissimilarity index was used for all pairwise combinations of assemblages between sample subgroups (n=3), based on presence/absence data. Multivariate dispersion was then calculated using the Jaccard dissimilarity index, which gauges the similarity and diversity of sample sets (Real and Vargas, 1996). To summarize the variation in community structure among subgroups, homogeneity swas assessed based on the mean distance of samples to their corresponding group centroid ("distance to centroid"), (Anderson et al., 2006). Statistical differences in mean centroid distances between sample groups were tested using PERMANOVA, using the adonis

function in the vegan package, with 999 permutations. Distance-based Redundancy Analysis (dbRDA) was finally used to explore the relationship between environmental parameters and relative abundance (Legendre and Andersson, 1999)

## 3 RESULTS

### 3.1 Stable water isotopes

Stable water isotopes illustrated the differences in water sources and flow paths between ponds and streams (Fig. 2). The evaporation line was well below the GMWL and the LMWL. Isotopic signatures from ponds confirmed evaporative fractionation, which was also reflected in the more depleted lc-excess values, particularly in summer and autumn. In comparison, effluent impacted streams showed limited variability and more depleted values, with moderately negative lc-excess as a result of concurrent mixing of multiple water sources, including local groundwater, surface runoff and effluent

discharge (Kuhlemann et al., 2020, 2021; Marx et al., 2021, 2023). Streams with no effluent discharge showed more variability, with seasonally more depleted signatures and overall more enriched lc-excess.

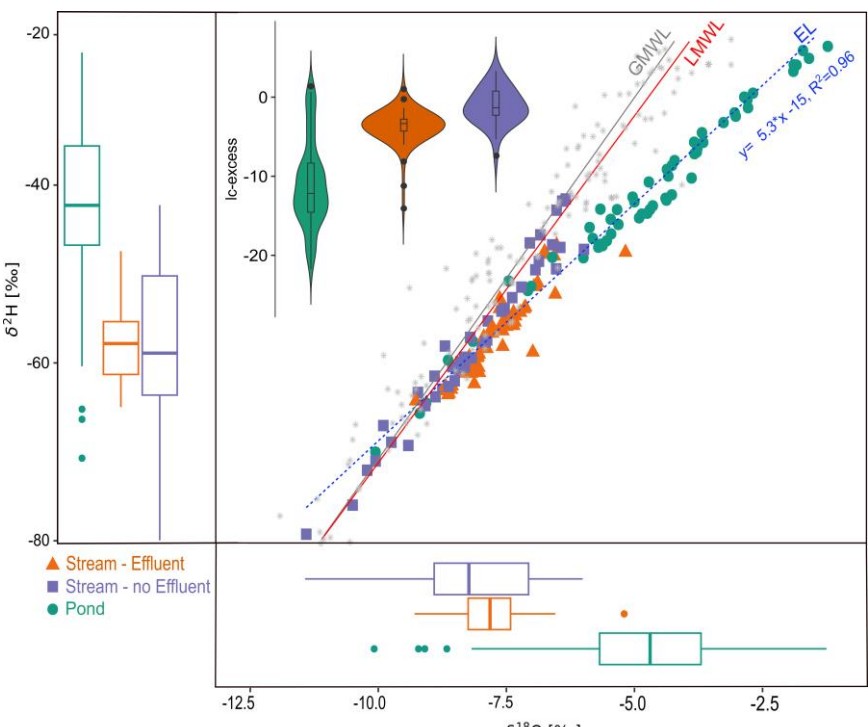

**Figure 2: Dual isotope plot (center) and box plots (left, bottom) showing the isotopic composition of the different surface water types, as well as lc- excess (inset). In boxplots, the horizontal line indicates the mean, whiskers indicate the range of values from minimum to maximum. Outliers are indicated as dots. Global meteoric water line (GMWL, grey), amount-weighted local meteoric water line**

**(LMLW, red)) from precipitation samples in Berlin-Steglitz and evaporation line (EL, black) are given for reference.**

Estimates of young water fractions (YWF; % of water younger than 2-3 months) in ponds were high, ranging between 50 to > 90% (Table 1). YWF in the effluent-impacted Panke and Erpe were characteristically lower (~10%), due to the dampening influence of wastewater (see also Figure S1). However, there was some variability in YWF estimates from samples of the
Panke river, with estimates of up to 50% (RSE = 0.5) in the most downstream restored section (Südpanke), which is highly influenced by weir management and urban surface runoff. YWF from the non-effluent impacted Wuhle ranged from 40% in the lower catchment to > 90% in the upper catchment, with greater uncertainties in the upper catchment (RSE ~1.0). This is in line with the variable influence and response of streamflow to seasonal precipitation and surface runoff, and groundwater in the upper and lower catchment, respectively.

**Table 1: Estimations of young water fractions for stream and pond sites from sine-wave fitting using IRLS** (Kirchner, 2016)**, including coefficient of determination ($R^2_{adj}$) and residual standard errors (RSE).**

| | $\delta^{18}O$ YWF | p-value | $R^2_{adj}$ | RSE |
|---|---|---|---|---|
| **Pond** | | | | |
| *Pianoteich* | 0.96 | <0.001 | 0.89 | 0.33 |
| *Orankesee* | 0.53 | <0.001 | 0.93 | 0.14 |
| *Obersee* | 0.77 | <0.001 | 0.76 | 0.44 |
| *Wuhleteich* | ~1.0 | 0.002 | 0.59 | 0.88 |
| **Streams - Effluent** | | | | |
| *Südpanke* | 0.54 | 0.01 | 0.47 | 0.49 |
| *Panke SP* | 0.32 | 0.057 | 0.29 | 0.42 |
| *Panke OL* | 0.17 | 0.31 | 0.04 | 0.36 |
| *Erpe* | 0.11 | 0.13 | 0.13 | 0.17 |
| *Erpe HG* | 0.04 | 0.83 | 0.14 | 0.22 |
| **Streams – no Effluent** | | | | |
| *Neue Wuhle* | 0.96 | 0.03 | 0.38 | 0.96 |
| *Wuhle* | 0.95 | 0.02 | 0.37 | 1.15 |
| *Wuhle OL* | 0.38 | 0.03 | 0.41 | 0.41 |

### 3.2 Hydrochemistry

Across all stream and pond sites, pH was slightly alkaline with means around 7.9 (sd = 0.6; Fig S1, Table S2). EC varied distinctly between sample groups (p<0.001), with higher EC in effluent impacted streams. Between ponds, EC also varied significantly, with higher EC observed in Obersee. (Fig S2, Table S2). Highest DIC concentrations were observed in the lower
catchment sites of the non-effluent impacted Wuhle (up to 85 mg/l), but on average were highest in effluent impacted Erpe and Panke and lowest in ponds. Dissolved oxygen in surface water varied significantly between subgroups (p=0.02), reflecting seasonal differences in local biological productivity. DO levels were highest in ponds and lowest in non-effluent impacted Wuhle in autumn and summer. Water temperatures ranged between 19- 23°C in summer and 1.5 – 6.8°C in winter.

Clear differences in major ion sources between stream and pond sites reflected the tributary nutrient loads from
effluent discharge into streams and the influence of dominant water sources (e.g. GW, surface runoff, precipitation) in ponds

and non-effluent impacted streams also seen in the isotopic signatures (see also Fig. S3). Most notably in the effluent impacted Erpe and Panke, levels of nitrate nitrogen ($NO_3N$), $SO_4$, Ca, K, Mg, P, S, Si were much higher compared to non-effluent impacted streams and ponds (see Table S3). Concentrations of Na and Cl were on average highest in effluent impacted stream, but also in Obersee. Elevated concentrations of Ca were observed in effluent impacted streams, as well as in Obersee (see Fig.

S3), which can be attributed to GW contributions from local aquifers, which tend to be Ca-rich in Berlin. Of all ponds, Obersee showed also higher levels of S and $SO_4$. Other trace metals such as Cu, Fe, Mn and Zn levels were negligible in all samples (< 0.001 mg/l). ANC was highest in effluent impacted streams due to overall increased nutrient loads. ANC showed large variability in the non-effluent impacted Wuhle and ponds, likely as a result of varying source water contributions (i.e. urban runoff, GW), also evident in the more variable stable isotope signatures.

### 3.3 Microbial community characterization

Of the 14034 bacterial ASVs, 48% (total 6693) could be assigned to a genus, of which ~ 80 % (total = 5465) were present in >10% of the samples. The final data set of bacteria consisted of 42 samples, with a total of 212 genera and a mean of 83 genera in each sample (range: 31-161). For diatoms/algae, 7125 ASVs were obtained, of which 65% (total = 4626) could be assigned

to a species and of those ~48% (total = 2188) were detected in >10% of the samples. The final dataset of diatoms/algae consisted of 46 samples, with a mean of 89 genera per sample (range: 5- 162). NMDS ordination of microbial communities shows an acceptable representation (stress <0.2 for both) (Fig 3). The community structures of bacteria and diatoms/algae were significantly different according to the NMDS (both p = 0.001), with communities of ponds being clearly differentiated from streams. Significant differences between stream and pond sites were confirmed by pairwise comparison of similarities

(ANOSIM, r=0.51, p = 0.001) for bacteria and diatoms/algae (ANOSIM, r= 0.46, p=0.001). Microbial communities in effluent and non-effluent impacted streams appear closely clustered in the ordination plot. However, the pairwise comparison indicated significant differences between effluent and non-effluent impacted streams in both bacteria (ANOSIM: r=0.53, p=0.001) and diatom/algae communities (ANOSIM: r=0.35, p=0.001). Seasonal differences in bacterial assemblage were significant in effluent impacted streams (p = 0.001), but negligible in other subgroups. For diatoms/algae, samples of streams visibly

converged in all seasons, suggesting similar seasonal community assemblage, although pairwise comparison (ANOSIM, r= 0.31, p=0.001) indicates some distinction between seasonal samples.

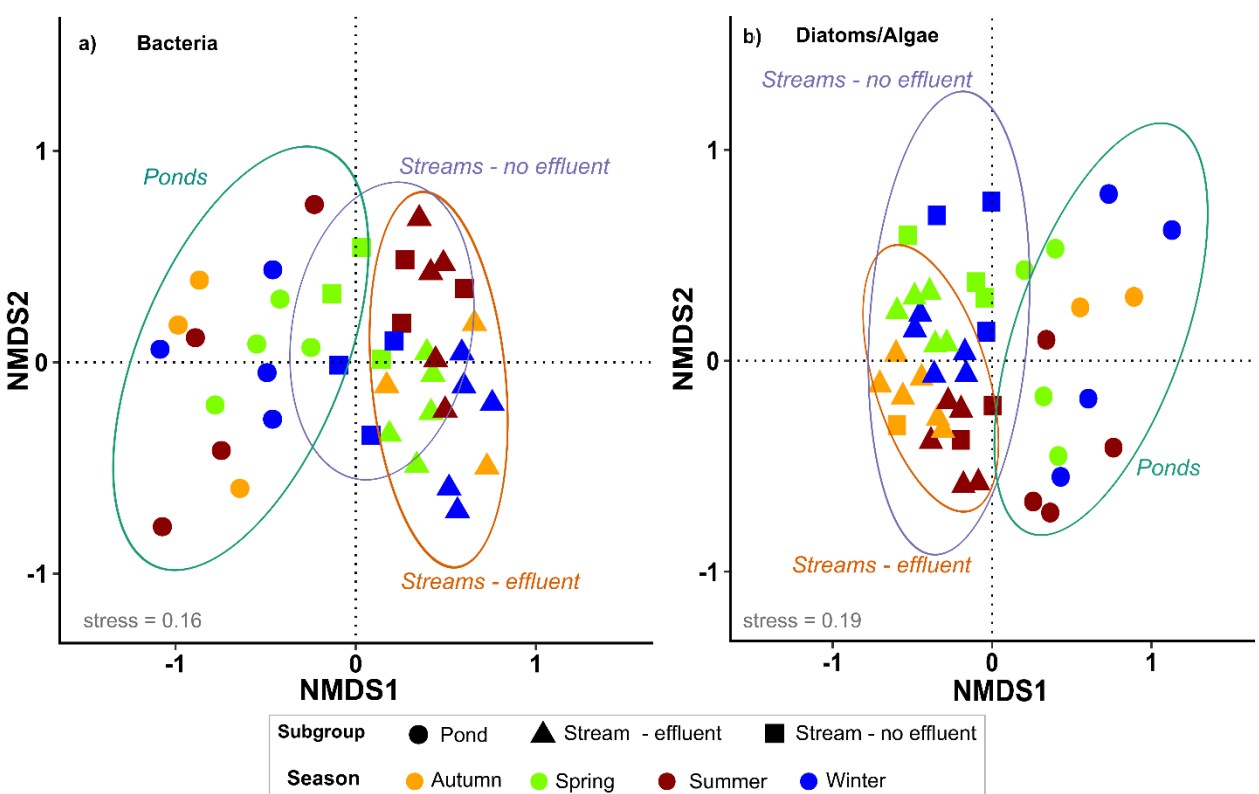

**Figure 3: Non-metric multidimensional scaling using Bray-Curtis dissimilarity index for a) bacteria (genus level, stress=0=16) and b) diatoms/algae (species level, stress=0.19) in urban streams (with/without effluent) and ponds. Seasonal samples are indicated by colours, shapes denote different subgroups. Ellipses indicate the 95% confidence interval for each subgroup.**

### 3.4 Alpha diversity

Compared to stream sites (mean: ~110), about ~50% fewer bacterial genera were found in ponds (mean: 55) Pairwise comparisons of genus richness indicated no difference among stream subgroups (p=0.72), but between pond and stream subgroups (p<0.001) (Fig. 4a). Most notably, in certain ponds and the non-effluent impacted Wuhle, there was a high relative abundance of Cyanobacteria (up to 35%). In ponds, they were most abundant in spring and lowest in autumn, while in the Wuhle they were most abundant in winter. Shannon diversity was not significantly different between ponds and effluent impacted streams (p=0.12), while diversity in the non-effluent impacted Wuhle clearly differed from ponds and the effluent impacted Erpe and Panke (Fig. 4d).

Diatom/algae species richness differed significantly among all sample subgroups (p =0.004) (Fig. 4b), with fewer species in ponds (mean: 75) compared to streams (mean: 103). Post-hoc pairwise comparison revealed significant differences between ponds and the effluent impacted Wuhle (p<0.001). Shannon diversity was similar (p=0.2) in all subgroups, suggesting no significant divergence in species diversity (Fig. 4e). Although the composition of diatom/algae communities in streams

appeared significant according to ANOSIM, the similarities in Shannon diversity suggest limited variation in species composition.

Macrophyte richness was low in all streams (Fig. 4c). Overall species richness did not show significant variability between subgroups across levels of naturalness (all p>0.1). Species richness was lowest in the non-effluent impacted Wuhle, with the identified taxa indicative of more eutrophic conditions, such as *Phalaris arundinacea, Iris pseudoacorus, Persicaria hydropiper* or *Sparganium emersum* (see Table S4 in the supplementary material for a complete list of species). These species represent emergent macrophytes (helophyte vegetation), which develop well in shallow zones of stagnant or slow-flowing waters and are also characteristic for nutrient rich water or sediments. The highest species richness was observed in the effluent-impacted Panke (n=12). Also, emergent macrophytes were dominant in the Panke, but beyond them, a greater diversity of submerged vegetation was recorded, with most of the species specific to nutrient-rich waters. Shannon diversity also showed no significant variability between subgroups, except in the non-effluent impacted Wuhle (p= 0.01) (Fig. 4f). However, this group only consists of 2 sample sites, thus may not be fully representative.

Pond sites exhibited greater consistency and comparable levels of macrophyte diversity. The diversity indices obtained from ponds were significantly higher than those from effluent-impacted streams, yet lower than those from the non-effluent impacted Wuhle. Additionally, the variability among individual ponds was comparatively lower. The findings indicated that, regardless of whether the sites were categorized as ponds, effluent, or non-effluent streams, the more anthropic locations were characterized by increased biodiversity. This phenomenon was less attributable to the anthropogenic alterations of the aquatic ecosystems and more related to the underdeveloped riparian or terrestrial vegetation linked to the transformation of the bank zone. Consequently, these modifications resulted in improved light availability (reduced shading), fostering enhanced growth and diversity of macrophytes.

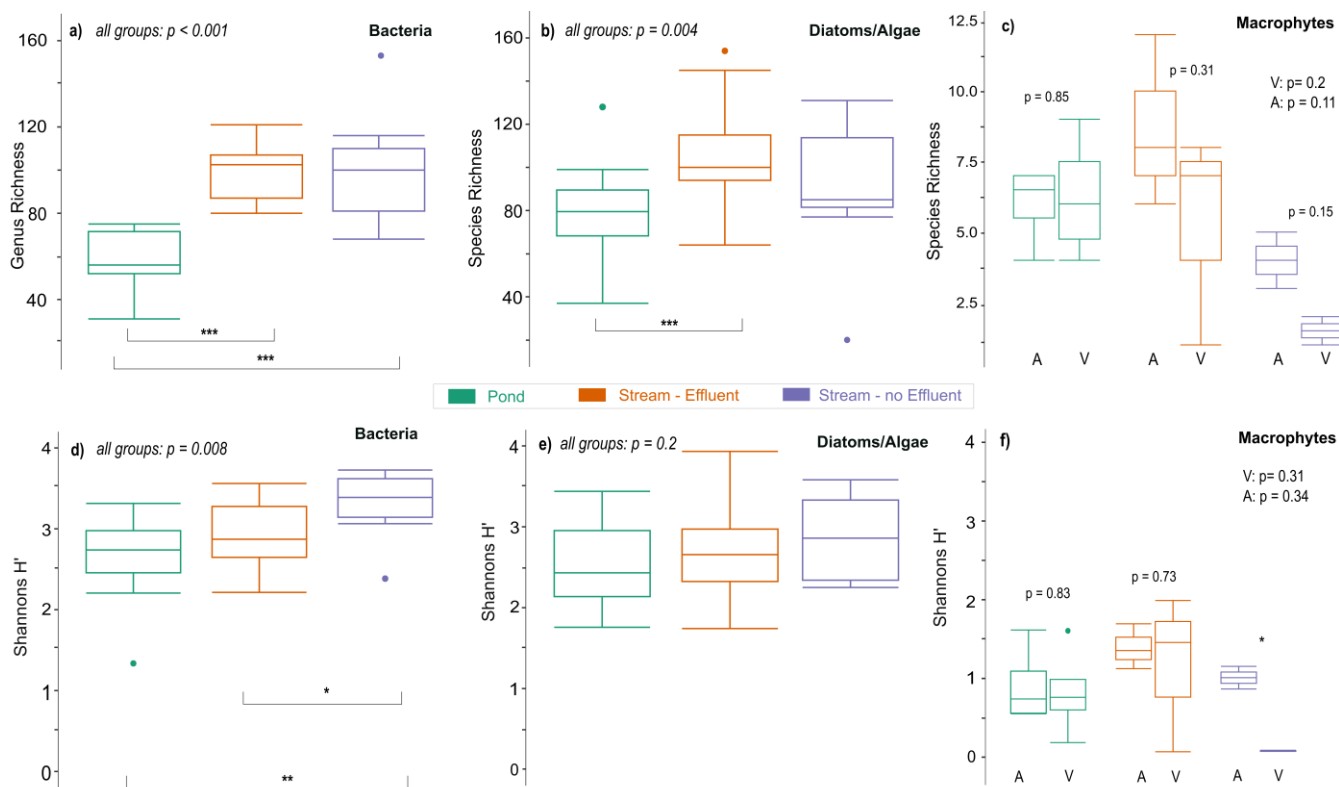

**Figure 4: a) Genus richness for bacteria, b) species richness for diatoms/algae and c) species richness for macrophytes. d)-f) Shannon diversity for bacteria, diatoms/algae and macrophytes. Horizontal lines in the box plots represent median values, dots indicate outliers. Colouring corresponds to sample subgroups. P-values denote significant differences in alpha diversity measured across all sample groups or between two individual groups. Statistical significance between two subgroups is indicated with asterisks at the 0.05 (\*), 0.01 (\*\*), 0.001 (\*\*\*) level. Macrophyte sampling sites are differentiated between vegetated (V) and artificial (A) sites.**

### 3.5 Beta diversity

Beta diversity dispersion, considering Bray Curtis dissimilarity, varied between all sample groups for bacteria (F=2.91, p = 0.001) and diatoms/algae (F= 3.02, p= 0.01) (Fig. 5). Overall, bacterial community dissimilarity was highest in ponds (0.41), compared to effluent and non-effluent impacted streams (0.30 and 0.28, respectively). Similarly, β-diversity of diatoms/algae was highest in ponds (0.51) compared to streams (0.45 in effluent impacted, 0.47 in non-effluent impacted) (see also Fig. S5). Considering all possible pairs of comparison between stream and pond sites, microbial dissimilarity was significant only between ponds and streams, but not between the two stream subgroups (bacteria: p = 0.96; diatoms/algae: 0.19), which contrasts the dissimilarity observed in the NMDS (Fig. 3). Partitioning total beta diversity into the relevant components, bacterial seasonal turnover ($\beta_{SIM}$) was highest for ponds (0.37) and lowest in the non-effluent Panke and Erpe (0.16), (see also Table S5). Similarly, diatom seasonal turnover was also highest in restored ponds (0.49) and lowest in the non-effluent impacted Wuhle (0.29).

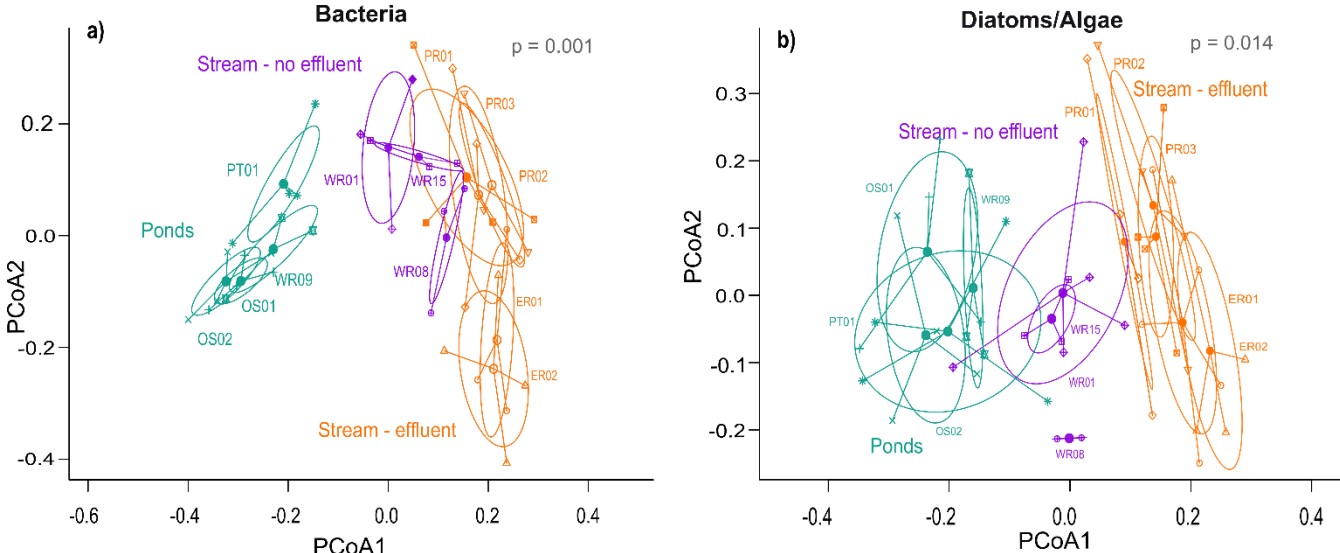

**Figure 5: Multivariate beta diversity dispersion of bacteria and diatoms/algae. Beta diversity is measured as the distance of samples to their group centroid (using Jaccard distances), shown here as the first two axes of a PCoA. On the PCoA, a change in site dispersion around the centroid indicates a change in beta diversity, while a change in the centroid location indicates species turnover. Symbols represent each stream or pond site, colours indicate the overall grouping.**

### 3.6 Influence of environmental variables on microbial and macrophyte communities

Distance-based RDA analysis revealed that almost 50% ($R^2 = 0.69$, $R^2_{adj} = 0.46$) of the variation of bacteria in streams and ponds ($R^2 = 0.83$, $R^2_{adj} = 0.53$) could be explained by physicochemical and hydrological parameters (see Table S6 for summary of RDA model results). The environmental variables most significant for the bacteria-environment relationship in streams were discharge (p=0.001), water temperature (p=0.001), DIC (p=0.007), DO (p=0.004), percentage of green space (p=0.003), YWF (p=0.005), Na (p=0.04), B (p=0.04), and EC (p=0.01) (Fig. 6a). Opposite relationships of bacterial community assemblage with nutrients and discharge variability separated bacterial communities between effluent and non-effluent impacted streams. YWF was positively correlated in non-effluent impacted Wuhle, where YWF were generally higher than in effluent-impacted Panke and Erpe (see also Table 1). Statistically, the first two RDA axis were significant (dbRDA1: p=0.001; dbRDA2: p=0.007), cumulatively explaining 38% of the total variation, with most of the variation occurring along the first axis (22.52%). Along axis 2, water temperature was a key driver of seasonal changes of bacterial communities, broadly separating summer and spring from winter and autumn samples, especially in the effluent impacted streams.

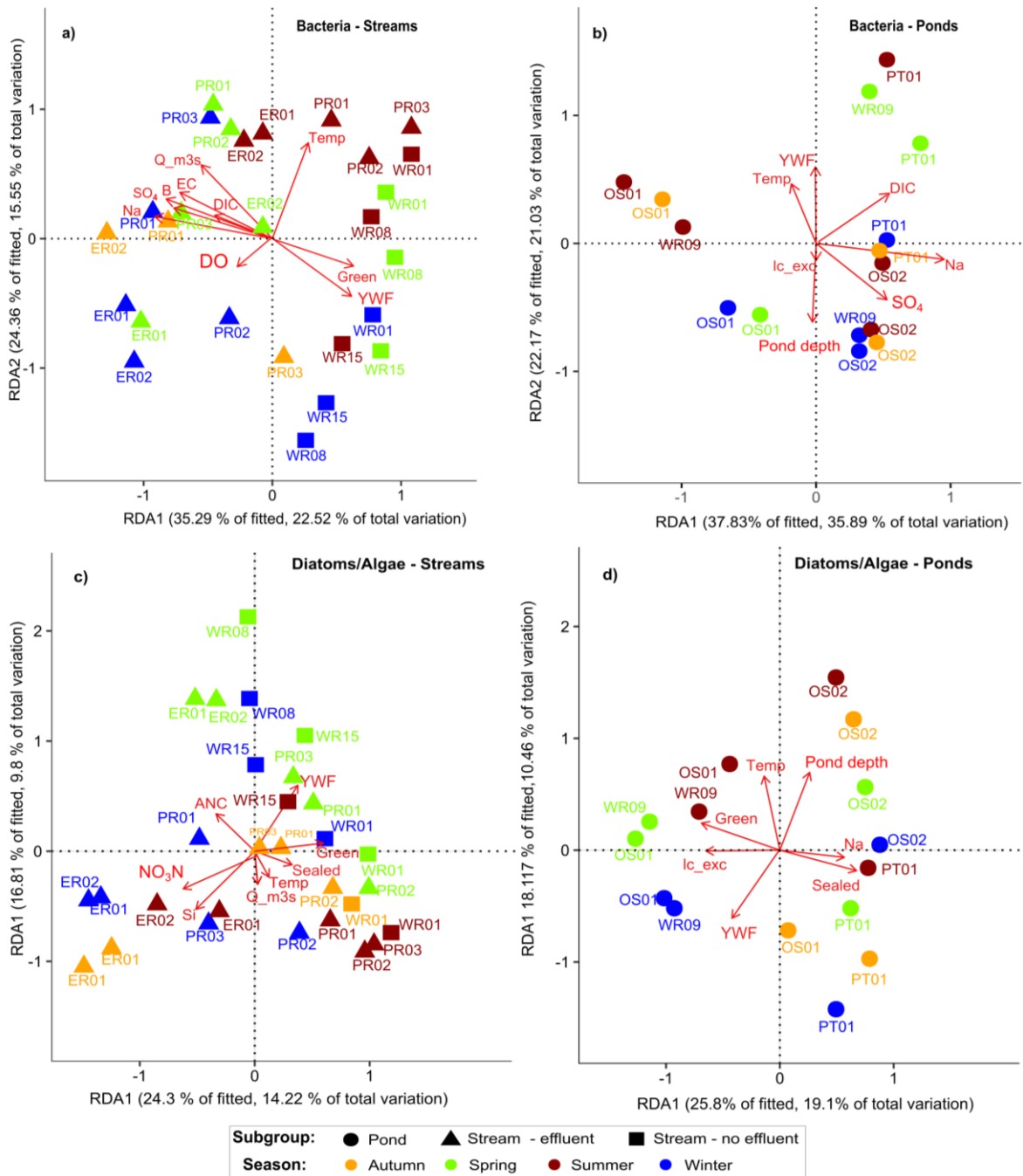

**Figure 6: Distance-based redundancy analysis (dbRDA) plot of distance based linear model (DBLM) of key environmental parameters fitted to the variation in bacteria and diatoms/algae for a) and c) streams (effluent and non-effluent impacted) and b) and d) ponds. Arrows represent direction of parameter effect, colours indicate sampling seasons, symbols denote subgroups for streams and pond sites.**

In ponds, pond depth (p=0.013), water temperature (p=0.019), DIC (p=0.001), YWF (p=0.003), Na (p=0.01), $SO_4$ (p=0.03) and lc-excess (p=0.01) were most significant (Fig. 6b). The first two axes were significant (dbRDA1: p=0.003: dbRDA2: 0.03), with ~ 36% of the variation occurring along the first axis (Fig. 6b). Na, DIC and $SO_4$ were positively correlated to

430 bacterial assemblage in ponds PT01 and OS02, in line with increased surface runoff reaching these ponds. YWF, water temperature, lc-excess and pond depth broadly separated seasonal samples, indicating a seasonal sensitivity to changing water levels and hydroclimate conditions, in line with the highly variable water isotope signatures and evaporative fractionation observed in ponds.

For diatoms/algae up to 30% of the variation in diatom/algae communities ($R^2 = 0.61$, $R^2_{adj} = 0.32$) in streams and ~ 20% in

ponds ($R^2 = 0.66$, $R^2_{adj} = 0.2$) could be constrained through surrounding landuse, hydrologic and physicochemical parameters. In streams, YWF (p=0.001), percentage of green space (p=0.001), percentage of sealed surfaces (p=0.009), discharge (p=0.004), $NO_3N$ (p=0.01), water temperature (p=0.003), ANC (p=0.002), and Si (p=0.03) (Fig. 6c) were most significant. The parameters that significantly constrained diatom communities were the % sealed surfaces (p=0.004), pond depth (p=0.005), Na (p=0.04), and YWF (p=0.002).

Due to only one sample date, pond and stream macrophytes were not divided by season or locality. Multivariate RDA showed, that up to 50% of macrophyte diversity ($R^2 = 0.73$, $R^2_{adj} = 0.49$) could be explained by concentrations of $NO_3N$ (p=0.001), P (p=0.004), Ca (p=0.001), alkalinity (p=0.008), pH (p=0.002), DO (p=0.007), YWF (p=0.01) and water levels (0.001). The results broadly match the patterns observed in microbial communities, with nutrient-rich waters in effluent-impacted streams accommodating specific species not found in the other sites. Compared to streams, the generally higher young water

contributions, and higher DO levels in ponds, appear to contribute to the similar levels of macrophyte diversity observed across all ponds. The exception wasOS-01, which had a high abundance of *Iris pseudoacorus L.* (swamp iris) and dense reedbeds (*Typa angustifolia L.*), which were less dominant or absent in the other ponds. Water level fluctuations appeared positively correlated with species diversity in ponds, but negatively in streams.

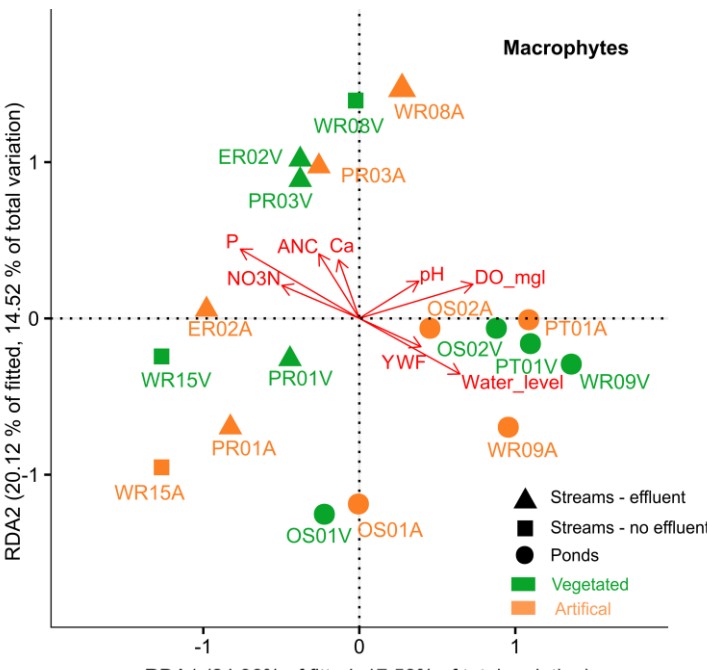

**Figure 7: Distance-based redundancy analysis (dbRDA) plot of DBLM (distance based linear model) of key environmental parameters fitted to the variation in relative abundance of macrophytes for streams and ponds. Symbols denote sample groups, colours denote artificialness (green – vegetated, orange – artificial). Arrows represent direction of parameter effect, colours indicate sampling seasons, symbols denote subgroups for streams and pond sites.**

## 4. DISCUSSION

### 4.1 Differences in microbial and macrophyte diversity

This study provides an integrated evaluation of the chemical and hydrological functioning of contrasting urban aquatic nature-based solutions. The observed differences in microbial diversity illustrate the influence of effluent on community composition in the Erpe and Panke, throughout the stream continuum (see also Kuhlemann et al., 2021; Marx et al., 2021; Warter et al., 2024). Multiple studies have already tied the effects of rapid urbanization and increasing connectivity between urban water bodies and sewer lines, to changes in natural freshwater bacterial communities across urban streams (Lee et al., 2020; McLellan et al., 2015; Numberger et al., 2022; Vignale et al., 2023). In cases where surface waters receive of water from different urban sources, such as urban storm drains and effluent discharge, their good ecological status remains constantly compromised the increased influx of nutrients and harmful substances (i.e. antibiotics, pharmaceuticals etc.) (Büttner et al., 2022; Lee et al., 2020; Reid et al., 2019). This is most evident in the abundance of highly specific bacterial communities associated with the human digestive system (i.e. *Romboutsia, Intestinibacter, Clostridium*), and N and P tolerant diatoms (i.e. *Nitzschia amphibia, Polytoma uvella*), throughout the whole downstream environment of the effluent-impacted Erpe and Panke. Similarly, the high

abundance of cyanobacteria during summer and autumn in the non-effluent impacted Wuhle, was likely a result of more eutrophic conditions, propagated by high urban runoff after storm events. In both cases, hydrochemistry, eDNA and stable water isotopes confirm the overarching fingerprint of urban-specific water sources on ecologic and hydrologic functioning. Based on our results, we would argue that the present ecological and hydrological conditions suggest a certain mismatch between the initial restoration goals and the current state of the surveyed stream sections. Overall, the findings build on a previous study by Warter al. (2024) in Berlin, confirming that the extensive influence of wastewater essentially limits the ecological potential, favouring the abundance of more nutrient-tolerant species, as seen in diatom as well as macrophyte species. The higher richness of macrophyte species in the effluent-dominated Erpe and Panke suggests that in urban environments, where organisms are affected by multiple pressures, the effluent input may be an important factor favouring the development of aquatic plants that prefer or thrive in high concentrations of nutrients (Gebler and Szoszkiewicz, 2022; Silva et al., 2020). This may create additional beneficial contributions to the purification of water from nutrients (Levi et al., 2015; Toerien and Toerien, 1985). Aquatic plants can also be an important element in shaping microbial communities, which is not only related to specific species of macrophytes (Wang et al., 2024), but also to their functional traits and diversity (Özbay, 2018; Zhu et al., 2021). Specific functional groups of microbes may also be associated with hydrophyte litter decomposition, which may further promote the development of microbial communities associated with reducing nutrient loads (Kumwimba et al., 2023; Toerien and Toerien, 1985). Still, the higher richness in effluent-impacted streams and the fact that bacteria show a more pronounced influence from effluent discharge than diatoms/algae may warrant further examination, particularly of sediment communities and composition as they constitute another reservoir of organic and inorganic compounds over time (Haller et al., 2009; Huang et al., 2020; Wang et al., 2023).

While nutrient levels and hydrologic variability were effective indicators of microbial and macrophyte communities in streams, pinpointing the key factors of microbial variability in ponds was more difficult. The observed low alpha diversity contained within urban ponds was at first surprising. This was generally in line with previous studies, which noted low biodiversity in urban ponds, including aquatic plants, macroinvertebrates and amphibians, due to the influence of water quality and urban landscape structure (Hamer and Parris, 2011; Oertli et al., 2023; Oertli and Parris, 2019). As the number of plant and animal species inhabiting a pond is strongly impacted by the surrounding landuse, a more urban matrix tends to limit the dispersal or exchange of species, often leading to biologic isolation of certain ponds (Oertli and Parris, 2019). Similarly, the exposure to urban water sources (i.e. storm runoff) and water level fluctuations, can create variable conditions that can promote greater species turnover (Siddha and Sahu, 2022), which is line with the observed high beta diversity and turnover in sampled ponds. The high evaporative enrichment also suggests a certain sensitivity to hydroclimate, with purely rainfed ponds such as Pianosee, Wuhleteich and Obersee, at risk of experiencing disproportionate water losses during hot and dry periods.

As urban ponds are often designed with predominately social or aesthetic ecosystem services in mind, the promotion and conservation of local native biodiversity through suitable, if less aesthetic habitat often falls behind (Bartrons et al., 2024; Cuenca-Cambronero et al., 2023; Oertli and Parris, 2019). The resulting biotic homogenization, due to a desire for uniform environmental conditions and functional designs thus creates habitat niches that will generally benefit only a limited number

of species (Hassall, 2014; Numberger et al., 2022). Beside physicochemical conditions, an observed lack of natural substrates and suitable plant beds, and the presence of steep concrete shorelines likely limited the establishment of more diverse aquatic species and macrophytes in observed ponds.

## 4.2 Future resilience of urban aquaNBS

Our study provides a novel inventory of hydrological and ecological functioning of aquaNBS within urban stream and ponds, while simultaneously challenging some of the restoration measures that were undertaken and their actual contribution to aquatic diversity. The disparity between the perceived effectiveness of restoration and the actual impact on aquatic diversity and habitat quality still presents a major issue for restoration efforts, especially in urban areas. This is due to a continued lack of post-hoc assessment of restoration objectives and a significant knowledge gap in maximising blue infrastructures as effective nature based solutions (Bartrons et al., 2024; Hilderbrand et al., 2023; Oertli and Parris, 2019; dos Reis Oliveira et al., 2020). To date, macroinvertebrate assessments are the most common methods used for assessing aquatic diversity (Durance and Ormerod, 2009; Fergus et al., 2023). However, challenges in the level of detection, taxonomic assignment, sampling technique, and representativeness, make it less viable for applications over extended temporal and spatial scales. As such, the multi-tracer approach constitutes a novel and innovative alternative to streamline assessments and long-term monitoring of in-stream restoration and associated effects on hydrological and ecological functioning.

Based on our results, we argue that to select appropriate aquaNBS and avoid ecosystem disservices (Lyytimäki and Sipilä, 2009), it is essential to first identify current and most relevant combinations of stressors that affect ecological conditions of streams – i.e. increased nutrients, urban runoff, surrounding landuse. For this, the integrated application of environmental tracers, such as stable isotopes and eDNA have proven useful in quantifying the biophysical interactions needed to characterize ecosystem functioning, which in turn can help to implement appropriate restoration measures. As the movement of water, matter and organisms, and interactions with the surrounding landscape is substantially different in urbanized systems (Oswald et al., 2023), prioritizing the scales on which stream restoration measures should have measurable impact (local, regional, or watershed) is critical for the viability of the NBS. Especially for aquatic nature-based solutions, upstream processes influence downstream conditions and local hydrology and the potential effects of hydroclimate changes will dictate the functioning and long term-benefits of the NBS. As such, water managers need to account for how the NBS might evolve over time with ongoing climate change and changing water availability.

Secondly, in riverine ecosystems it is crucial not to disregard natural system complexity in favour of simplistic and static minimum flow designations, especially considering the effects climate change and urban stormwater (Acreman et al., 2014; Arthington et al., 2006). In the case of the non-effluent impacted Wuhle, an increasing trend towards intermittency in recent years and a more variable flow regime, coincide with an observed limited macrophyte diversity, lower water quality and high abundance of cyanobacteria. This suggests that past restoration measures may not be able to counteract the ongoing loss of hydrological connectivity, making it unlikely to improve ecological conditions in future. As such, projections of longer

droughts followed by intense rainfall are likely to reduce the efficacy of NBS that depend on vegetation-surface water interactions to improve water quality and streamflow regimes (Fletcher et al., 2024). In drought-prone regions, effluent discharge has already become an increasingly important water source for sustaining such declining stream flow regimes (Luthy et al., 2015), stabilizing discharge variability and keeping baseflows high even in times of drought (Lawrence et al., 2014; Plumlee et al., 2012). However, there are noted caveats to consider for aquatic habitats, with potentially long-term effects on ecology, chemistry and flow dynamics, which are still not well established (Büttner et al., 2022). As an alternative, integrating aquaNBS in the form of restored wetlands or enhanced river channel restoration, to engage hyporheic filtration processes or slow-rate soil infiltration, would more actively support the removal of pollutants in effluents before they are discharged into stream ecosystems (Büttner et al., 2022; Cross et al., 2021; Stefanakis, 2019).

Due to the pervasive influence of urbanization on water quality in the studied ponds, the contributions towards aquatic diversity ultimately appears to be quite limited. Indeed, a caveat to this conclusion is that we did not consider terrestrial vegetation or higher-order aquatic organisms, such as amphibians, macroinvertebrates or fish, as this was outside the scope of this study. However, diversity patterns of bacteria and diatom/algae communities already provided an initial assessment of ecological conditions, which could be clearly linked to hydrology, water quality and urban influences. This is not to say that urban waterbodies cannot support freshwater biodiversity at all or provide storm water storage, nutrient processing or recreation opportunities (Fletcher et al., 2024; Hassall, 2014). Indeed, there are many examples of threatened and endangered species finding refuge in urban parks, wetlands and ponds. Examples include populations of macroinvertebrates (Hill et al., 2016), damsel- and dragonflies (Goertzen and Suhling, 2015) and amphibians (Holtmann et al., 2017). However, the proximity to human activities, limited green space, and increasing exposure to urbanization and climate effects, clearly affected microbial diversity in studied ponds. Here we argue that pond design must consider not only physical requirements but also the physiochemical habitat quality and biological functions in order to reconcile ecologic with aesthetic or social ecosystem services (Bartrons et al., 2024).

In future, to improve the success and longevity of restoration projects, clear expectations need to be set for the provision of ecosystem services and desired contributions to biodiversity (Bartrons et al., 2024; Oertli and Parris, 2019). The focus on trade-offs between societal and environmental issues, such as flood control, recreation or carbon storage, often leaves ecological and biodiversity aspects behind or considered separately, when in fact they are deeply connected and often share the same drivers (Seddon et al., 2020). More importantly, expectations may differ between protecting relatively unimpacted natural streams and restoring already degraded streams. For example, restoration goals in heavily degraded streams may aim simply create ecosystem structure and functions that provide certain services and social benefits (i.e. recreation, rainwater retention), rather than completely restoring natural conditions (Fletcher et al., 2024). Such highly modified urban water bodies may then still support water storage, nutrient processing and recreation or the provision space and amenities for residents, but fail to significantly contribute to biodiversity (Seddon et al., 2020).

Despite certain caveats to urban stream restoration and urban aquatic nature-based solutions, well designed approaches that match relevant biophysical and ecological objectives and restoration goals by considering basic microbial processes,

physiochemistry and hydrological functioning of aquatic environments, as well as relevant spatio-temporal scales, can greatly support climate change mitigation whilst also provide key benefits to people and the natural environment. Due to the important functions they perform, macrophytes can be a vital element in restoring and shaping the appropriate structure of aquatic ecosystems (Suren, 2009). This can also increase the diversity of all organisms associated with its occurrence and the general diversity of freshwaters (Fu et al., 2024; Paice et al., 2016). However, to achieve positive ecological outcomes, including enhanced water quality and rehabilitation of native populations, site-specific management and restoration strategies tailored to biotic and abiotic conditions (i.e. habitat, streamflow permanence) are crucial.

## 5 CONCLUSIONS

Our analysis of urban aquaNBS in streams and ponds within the city of Berlin demonstrated the influence of anthropogenic and environmental stressors on hydrologic variability, water chemistry, and microbial diversity. Despite the extensive restoration measures undertaken in all streams within the last decade, urbanization remains a multifaceted driver that impacts aquaNBS. Our use of an integrated multi-tracer approach proved to be highly insightful in qualitatively assessing microbial and macrophyte diversity and relationships with environmental parameters, addressing questions related to the effectiveness of restoration measures and aquaNBS in contributing to local aquatic diversity. Our research has shown that i) microbial and macrophyte diversity is heavily influenced by urban water sources and ecological potential may be severely limited in effluent impacted streams; ii) microbial diversity was lowest in urban ponds, with high abundance of cyanobacteria and low water quality; iii) high abundance of cyanobacteria and low macrophyte richness and diversity in non-effluent impacted streams suggests low contributions to local aquatic diversity despite extensive restoration; iv) microbial processes should be considered in future restoration efforts to create balanced ecosystems and improve ecological status of degraded ecosystems; v) the still limited research on restoration effects on biodiversity warrants further research to enhance climate resilience of urban waterbodies.

**Acknowledgements:**

We thankfully acknowledge Franziska Schmidt from the IGB Isotope Lab for the isotope analysis. We also are very grateful to colleagues from the Chemical Analytics and Biogeochemistry Lab of IGB for water chemistry analysis, in particular Claudia Schmalsch, Thomas Rossoll, and Marvin Sens. For support with the metabarcoding analysis of eDNA samples we thankfully acknowledge Sarah Sparmann, Susan Mbedi and Stephanie Neitzel. We also thank Jan Christopher for help with sampling.

**CRediT author contribution statement:**

Conceptualization: DT, CS, MM

Methodology: MMW, DT, CS, MM, TG, DG

Investigation: MMW, DG

Formal Analysis: MMW, DG

Data Curation: MMW, DG

Writing-Original Draft: MMW

Writing – Review & Editing: DT, CS, MM, TG, KV, DG

Visualization: MMW

Supervision: DT, CS

Funding acquisition: DT, MM, KV

**Data Availability:**

Data will be made available on request. Metadata can be found on the Freshwater Research and Ecology Database (FRED) at doi: 10.18728/igb-fred-939.0. Climate data was publicly available from the German Weather Service (DWD, 2023). Stream discharge and water levels are publicly available from the Berlin Senate (SenUVK, 2024).

**Funding:**

This study was funded through BiodivRestore for the Binatur project (BMBF No. 16WL015). DT also received funding through the Einstein Research Unit "Climate and Water under Change" from the Einstein Foundation Berlin and Berlin University alliance (grant no. ERU-2020-609) and through the German Research Foundation (DFG) as part of the Research Training Group "Urban Water Interfaces" (UWI, GRK2032/2). Research was also partially funded by the Einstein Stiftung Berlin, MOSAIC project Grant/Award Number: EVF-2018-425. MM acknowledges funding from the German Federal

Ministry of Education and Research (BMBF, No: 033W034A). DG was also funded through the Binatur project, which is partially financed by the National Science Centre (Poland) UMO-2021/03/Y/NZ8/00100.

The authors declare that they have no conflict of interest.

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
