# Peer review of "Understanding ecohydrology and biodiversity in aquatic nature-based solutions in urban streams and ponds through an integrative multi-tracer approach"

_EGUsphere, 2024_

## Author Response (AR1)

**Response to Referee Comment 1:**

We thank the reviewer for their careful review and appreciate the positive feedback on our study and suggestions for improvement. We believe addressing the comments will be straightforward and strengthen the paper and provide clarity where needed, to better deliver the key message we are trying to convey. Below, we respond to the specific comments point by point and provide clarification where needed. Line numbers refer to the tracked-changes manuscript.

Sincerely,

Dr. Maria Magdalena Warter (on behalf of all co-authors)
* * *
The authors collected water samples monthly from urban streams and ponds around the City of Berlin over a one-year period. Some streams received treated effluent from a wastewater treatment plant, and some did not. The samples were analyzed for stable isotopes of hydrogen and oxygen in water and a suite of physicochemical parameters (e.g., major ions). Samples were also collected for environmental DNA analysis, but only seasonally (n=4). In September of the study period, macrophyte (aquatic vegetation) surveys were carried out at two sites per each stream and pond site. Spatial and seasonal patterns in physiochemical parameters were assessed and spatial patterns in microbial diversity and community structures were assessed. Ultimately, the authors were interested in identifying the spatial and seasonal drivers of microbial variability and evaluating the significance of their results for assessing the benefits (and efficiency) of aquatic nature-based solutions (aquaNBS). The authors found that both urban streams and ponds have low macrophyte diversity, suggesting that current aquaNBS may not effectively support aquatic biodiversity. Indeed, the authors showed that microbial diversity is still strongly influenced by urban water sources (e.g., treated effluent, impervious surface runoff) despite implementation of aquaNBS. Effluent-impacted streams had a high abundance of bacterial communities specific to the human digestive system. Urban ponds had the lowest microbial diversity and were sensitive to hydroclimatic shifts. Ultimately, the authors advocate for implementing this multiple-tracer approach to assess the performance of aquaNBS across scales.

General Comments:

This is a high-quality study. The number of sites and number of tracers (that holistically characterize the ecosystem) makes up for the shorter time frame of the study. The integration of isotope, hydrochemistry, macrophyte, and microbial results is well done and relatively easy to follow. There are a few places where the methods could use some clarity (see suggestions below), but they do not detract at all from the overall understanding of what was done.

I think the scientific importance of this paper is high. While some of the results may not be surprising, I think the integration of multiple tracers is quite novel, especially in the context of aquatic ecosystem restoration. I think the paper would benefit from the authors leaning a bit more into the discussion of restoration goals. See my suggestions on this below. This could also include some discussion of the logistical constraints (e.g., time, cost, expertise) involved in implementing a multi-tracer approach more broadly. Ecological restoration is often implemented by municipalities, often with support from consultants. What kind of partnerships could be developed to make this approach accessible to all involved?

The paper is well-written with relatively few grammatical and linguistic errors. The figures are high quality, and all seem well placed in the main manuscript. I have provided some suggestions below on how to improve the clarity of some of the more data heavy figures.

** Thank you for this positive evaluation of our manuscript!

We appreciate the suggestion to lean more into the restoration aspect and also discuss some logistical aspects of the multi-tracer approach. We will aim to include some aspects regarding feasibility of the multi-tracer for ecological restoration purposes and the kind of potential partnerships this might require. We believe there is great potential to apply this tracer approach for a wide range of ecological questions and would welcome collaborative partnerships between municipalities and research institutes to increase the performance and resilience of nature-based solutions in future.

Specific Comments (and some Technical Corrections):

Lines 97 to 100 are repetitive.

** We have removed the redundant sentence (L110-111).

The research objectives do not mention macrophytes, despite them being discussed in the methods.

** We have amended the objectives to also mention macrophytes (L114-116).

Lines 72 to 73 are unclear. How do macrophytes drive competition?

**We changed the inadequate wording in this section (L76-78). What we meant was that macrophytes are competitors for other autotrophs, especially phytoplankton. But at the same time, they also provide habitat, shelter and food for other organisms, thus significantly contributing to the overall biodiversity of aquatic ecosystems.

Lines 80 to 82 seem important. Can you expand on how these previous studies used macrophytes and microbial communities as indicators of ecosystem health and functioning?

** We revised and expanded this section to include previous studies where macrophytes and microbial communities have been used as bioindicators (L 88-94).

The wording on lines 83 to 85 does not make sense to me. Can this long sentence be broken up?

** We broke the sentence in two (L93).

On lines 97 to 100 there are two aim statements that seem repetitive. I recommend combining them into one statement.

** The redundant statement has been deleted (L 110).

The research objectives do not mention macrophytes which are clearly an important part of this study and presented in the methods, results and discussion. I recommend integrating them into objectives one and/or two.

**See above. We rephrased objectives to also specifically mention macrophytes (L114-116).

On line 103 aquatic nature-based solution efficiency is mentioned. This term should be defined earlier in the introduction.

**A more thorough definition of aquatic nature-based solutions and relevant examples has been included in the first paragraph (L 46-48).

Double check the whole paper for missing or problematic punctuation. For example, on line 108 there is a missing period.

**We checked all punctuation throughout the manuscript. Punctuation in L122 has been changed.

On line 132, the authors say that the word 'restored' will not be used for the remainder of the paper because all of the sites are restored. But then the word is included in the subsequent subtitles.

**We removed the term "restored" from the subtitles in the manuscript.

What is the significance of the gray area in figure 1a. Why are the upstream portions of the Panke, Wuhle, and Erpe catchments not included in this gray area? And in figure 1c and 1d, why are those upstream portions not included.

**There is no particular significance of the grey area, other than highlighting this as the main city boundaries of Berlin in reference to the catchment boundaries. The separate coloring of the respective catchments was then chosen to illustrate that only parts of each catchment actually lie within the city limits of Berlin. As sampling points only primarily located within the city, we chose then to only show land use and surface sealing within the Berlin area.

On lines 204 and 205, can the authors explain how the natural and disturbed sites for the macrophyte surveys were chosen? Was it a random selection? How did the authors avoid bias in the choice of sites?

** Study sites were selected using a stratified random sampling approach. Site stratification considered the level of naturalness and alteration and the most and least altered quadrants within each site along a 100m stretch. We also considered variability in the type of aquatic nature-based solution at each site, to represent different kinds of implementations.

The paper by Szoszkiewicz et al. 2025 explains the approach in more details. The reference has been updated and the text amended (L248-252).

On lines 216 to 218, can the authors provide a reference and/or an equation for Shannon's diversity index.

**The reference to a paper by Hill et al., (2003), where the Shannon's diversity index, incl. it's calculation/equation, is discussed, has been added to the manuscript (L267).

In section 2.6, there are a number of statistical tests and methods mentioned, but it is not clear if these are traditional or novel (or somewhere in between) methods for this particular application. I think this could be addressed through a few references indicating previous studies that have applied these methods in a similar context. Also, what software was used for the analyses? This should be stated and referenced.

** Most of the statistical tests and methods are widely applied for the assessment of ecological data/diversity patterns and to determine statistical differences between sample groups. We provided additional references, where needed, to illustrate similar applications by previous studies.

For all analyses, R studio (v 4.2.3) was used. This has been added in the text (L258). Relevant R packages have also been referenced.

I think the word 'spatial' can be removed from line 228. If it is left in, it gives the impression that within group spatial variability is shown as well.

**This has been removed (L282).

On line 229, 'strong evaporative' does not make sense.

** We removed the word strong and changed to "evaporative fractionation" (L283).

On line 232 how do the authors know that there are multiple sources of water contributing to this stream. I think a reference to prior work would be helpful here.

**We added references to prior work (i.e. Marx et al., 2022, 2023, Kuhlemann et al., 2021, 2020) where this has been shown (L287).

On lines 234 to 240 can the authors provide an estimate of the uncertainty in the young water fraction. Does the frequency of sampling impact the estimates?

** RSE is included in Table 1 for all site. We included references to RSE values in the text as well to give context to the young water estimates and potential uncertainties.

Based on previous studies, sampling frequency does impact estimates – in general, the greater the sampling frequency (i.e. daily), the lower the uncertainty and more reliable the young water estimates are. However, in water bodies were isotopic variability is generally more damped, monthly sampling can be sufficient to give an approximation of young water fractions.

On line 274 DO is defined a second time. See line 186.

** We removed the second definition (L333).

On lines 282 to 283 if groundwater contributions vary, does that mean that calcium concentrations would vary? The authors describe Ca concentrations as 'elevated'. This sentence needs clarity.

** In general, GW contributions may vary seasonally – being higher in winter and lower in summer - due to water level fluctuations, but overall in these streams Ca concentrations are overall more elevated compared to streams without significant GW contributions.

We rephrased this sentence to be more concise and removed the word "varying" to avoid confusion (L342).

On line 290 percentages would be easier to interpret.

**We added percentage values for easier interpretation (L349-353).

On line 334 what is meant by stability of diversity?

**We used inadequate wording. What we meant was that pond sites exhibited greater consistency in species richness and comparable levels of macrophyte diversity.

We changed this in the manuscript (L395).

On lines 336-338 what is meant by 'more modified'? Is this referring to the artificial sites?

**In the context of the macrophyte assessment, this refers to the more artificial sites, with less vegetation surrounding the stream banks.

We changed this to the word "anthropic" instead (L 398).

In figure 6 can the headings for each panel indicate bacteria or diatoms and streams or ponds. This would be instead of having bacteria and diatoms only on panels a and c and streams and ponds written in the lower left corner.

**We amended the headings in each panel.

It is not clear in figure 6 what the different symbols mean. For example, in the streams panels could a legend be added for effluent and non-effluent impacted? And what are the different subgroups for the pond sites?

** We changed the symbols in all plots to match the symbols in Figure 3. We added a legend to the plot to show the different subgroups. We also amended the colors for each season, to match the colors in Fig. 3.

On line 369 and line 392, instead of 'green space' can the authors say 'percentage of green space in the catchment' (assuming this is correct)? The same applies to 'sealed surfaces' (and is applied on line 394).

** We changed "green space" into "percentage of greenspace in the catchment" and "percentage of sealed surfaces in the catchment" (L437).

On line 396 it is not clear what the authors mean by 'pond and stream macrophytes were assessed together and not separated'.

**We meant to say that because we only sampled macrophytes on one date, we did not separate them by season or locality for the analysis, like we did for the microbial DNA. This would have resulted in

too few datapoints and potentially less meaningful comparisons. Therefore, stream and pond macrophyte assessments were analyzed together.

We amended the sentence in L467.

Why is there a greater precipitation influence in ponds than streams (lines 400 and 401)?

\*\*In general, higher young water fractions indicate a greater influence of recent water, i.e. precipitation. Our analyses showed that especially in ponds, precipitation was the main water source, thus resulting in the higher young water fractions compared to some of the stream locations, where other water sources, such as groundwater or effluent discharge, are present.

The sentence that runs from line 400 to 404 is too long and doesn't completely make sense to me. At the beginning, ponds and stream are being compared, but then the sentence transitions to talking about macrophyte diversity across ponds. Can this sentence be broken up for clarity?

\*\*We split this sentence into two parts (L471-476).

On lines 420 to 422, is the bacterial community observed in the effluent impacted stream influenced by the harmful substances and nutrients? I am trying to understand if this sentence is directly or indirectly related to the information right before it.

\*\*Yes, the sentence only indirectly relates to the previous statement about the freshwater bacterial communities in urban streams.

The bacterial composition we observed in stream sites in Berlin, could be directly related to the impact of effluent discharge. However, in urban freshwaters the inflow of potentially harmful substances from surface runoff or point pollution sources does have effects on the overall ecological status and water quality, thus potentially also propagating certain bacterial communities.

We made some changes to the sentence to be more concise (L492-495).

On line 426, what is meant by low water quality?

\*\*Here we refer to the generally more eutrophic conditions in this stream during certain times, where low O2, high nitrogen was observed. We deleted "low water quality" to avoid confusion (L499).

On line 429 the authors refer to a mismatch between the initial restoration goals and restoration measures undertaken. Do the authors know with certainty that the restoration goals included a bacterial community that was reflective of pre disturbance conditions?

\*\*Unfortunately, we don't really have a (bacterial) reference to pre-disturbance conditions. However, when comparing our overall results of water quality, microbial patterns and macrophyte diversity to the intended restoration goals presented by the Berlin Senate in their public outline for each stream (see here: https://www.berlin.de/sen/uvk/umwelt/wasser-und-geologie/europaeische-wasserrahmenrichtlinie/berlin/erpe/ or here https://www.berlin.de/sen/uvk/umwelt/wasser-und-geologie/europaeische-wasserrahmenrichtlinie/berlin/wuhle/ ) – we would argue that a certain mismatch between the initial intentions and the current state of the restored streams has emerged. The increasing intermittency in one of the stream sites and the pervasive influence of the effluent discharge remain key aspects that impact water quality and microbial dynamics, despite extensive physical measures that were undertaken.

We slightly rephrased this section, to make this clearer and indicate that this is purely an interpretation by the authors rather than a quantitative comparison to pre-disturbance conditions (L501-505).

There appears to be an extra parenthesis on line 439.

\*\*We removed the extra parenthesis (L515).

Line 455 and 456: are urban ponds not designed with stormwater management as the main goal?

** Not necessarily. In the urban context, it may be implicit that ponds will arguably also function as storm water retention ponds. However, when designed with biodiversity aspects in mind, urban storm water inflow may be less desired due to the inflow of diverse contaminants that may impact water quality and habitat. Depending on the intended function (aesthetic, storm water retention pond, biodiversity hot spot, swimming), ponds may be designed differently, exhibiting different structural and water quality conditions, which directly affects potential habitat conditions and aquatic diversity.

Line 463: missing e in i.e.

**The relevant sentence has been deleted.

I am having a hard time extracting the key point from the last paragraph in section 4.1 (lines 462 to 467). In particular the first couple of lines which refer to 'successfully implemented' and 'contributions to biodiversity and water quality ensured' make me think twice. Are the authors advocating for this type of coupled isotope/hydrochemistry/microbial assessment to be implemented more broadly across NBS projects?

**Yes, based on the experiences from this study, we would advocate for this kind of integrated tracer approach to be widely applied in ecosystem restoration and NBS projects. Especially in the context of before/after assessments and long-term monitoring of successful restoration or NBS implementation, the tracer approach constitutes and effective and inexpensive way to assess water quality and biodiversity improvements. We believe that the current lack of monitoring frameworks and approaches of NBS and restoration projects could be remedied by adopting this kind of tracer approach as a way forward to monitor ecological and hydrological changes over time and assess actual contributions to restoration goals.

We moved some of the information from this paragraph into the next section where this is discussed, to avoid redundancy (L 549-554).

Is it a feasible approach to see if the cumulative effects of multiple NBS have a positive ecohydrological effect at the watershed scale?

**Yes, we believe it is feasible to also assess cumulative effects. Especially NBS along rivers and streams generally do not function in isolation but are affected by upstream catchment behavior and in turn also affect downstream processes. Since the tracer approach functions across extended temporal as well as spatial scales, it can be quite easily applied across the watershed scale. In addition, tracer-based models – either using stable isotopes or eDNA data, can be applied to explore larger watershed dynamics.

This goes back to my earlier comments about restoration goals. Should we expect a stream or pond that is integrated into the urban mosaic (and especially one that receives WWTP effluent) to regain any semblance of its pre-disturbance ecohydrological characteristics? If yes, then maybe this multi-method assessment is a good idea. If not, then why go to the trouble? The example on lines 512 to 514 gets at this type of decision-making, but I think it could be explored more.

**The most common misconception regarding NBS and stream restoration is that simple physical changes to the stream environment, such as changes in channel structure or morphology, will invite a return of pre-disturbance species. However, based on the results of this study, if a stream or pond still receives a large amount of WWTP effluent, the aquatic diversity and water quality conditions will inherently be limited. However, if the goal of restoration was to provide additional biologic cleaning and retention of nutrients (i.e. through biological filtration), it would make sense to trace the effectiveness of these processes through the multi-tracer approach. Otherwise, if the goal was to purely

create a more natural stream environment to support flood retention, micro climate regulation, or recreation, then the multi-tracer method need not be applied.

Could the authors expand on what is written on lines 471 to 473? Are there studies that review the most common stream and/or pond restoration objectives? Are 'increased aquatic diversity' and 'increased habitat quality' common goals? If so, how are they usually assessed?

**A study by Oertli et al., 2019 as well as Bartrons et al., 2024 review performance of urban ponds and the potential challenges. Oertli et al., 2023 have studied the functionality of ornamental ponds in cities and found that in most urban ponds, aesthetic enjoyment is the main ecosystem service targeted. The study actually advocates for a more holistic design that considers a multifunctionality of ponds, solving not only societal challenges but also contributing to biodiversity and water management challenges. In Oertli et al.. 2023, biodiversity and environmental variables were actually assessed across multiple ponds through macroinvertebrate assessments and water quality (i.e. pH, temperature), pond morphometry (i.e. surface area, depth, drawdown height), pond vegetation, and other features (i.e. presence of fish or ducks, trees, pond substrate).

In general, macroinvertebrate assessments are the most common method used for assessments of in-stream diversity. However, the challenges in level of detection, taxonomic assignment, sampling technique, representativeness, make it less viable for applications over extended temporal and spatial scales. Here, the multi-tracer approach is much more efficient and can be extended to cover a range of species, incl. macroinvertebrates.

We added additional explanations to the text (L545-555).

In the introduction (e.g., lines 45-47), goals associated with water quantity and heat are mentioned. Vegetation, habitat, and societal benefits are mentioned in the next paragraph. Perhaps some of these references could be revisited in this part of the discussion.

**We included some of the references from the introduction in the discussion

The paragraph starting on line 479 is very interesting. So ultimately hydrology will dictate the long-term benefits of NBS.

**Yes, especially for aquatic NBS. Any changes to the hydrological dynamics, i.e. streamflow permanence, water levels, flooding, will affect how the aquatic NBS evolves over time with ongoing climate change and its contribution to various ecosystem services.

We added a statement highlighting this (L560-565).

**Response to Referee Comment #2:**

We thank the reviewer for their careful review and suggestions. We appreciate the positive feedback and interest in our study. We believe addressing the comments will strengthen the paper and provide clarity where needed to better deliver the key messages we are trying to convey. Below, we address all major and minor comments by the reviewer point by point and provide clarification where needed. Line numbers refer to the tracked-changes manuscript.

Sincerely,
Dr. Maria Magdalena Warter (on behalf of all co-authors)

This paper applies a multi-tracer approach to assess ecological condition of four restored urban ponds and several sites along three restored urban streams around Berlin, Germany. The authors sampled a suite of hydrochemical parameters and stable water isotopes; macrophytes; and microbial communities; and the authors assess differences in these parameters between sites in order to evaluate the realized impacts of aquatic nature-based solutions (aquaNBS) restoration efforts in this urban landscape. The authors find meaningful and significant degradation of biodiversity in the pond sites and a restored stream influenced by effluent, and conclude that the benefits of restoration efforts may be limited by urbanization, which has important implications for site selection and restoration strategies in urban settings. The science is robust, interesting, and important. The conclusions are well-supported by the study design and results. The manuscript is generally well-written and concise, though a number of text errors are a bit distracting and should be resolved. My main suggestion to improve the manuscript would be to add more contextual and technical details to the Methods section and perhaps a bit more interpretation to the Results section. Connecting some of the dots for readers will open this important study up to a broader audience. I've offered some suggestions below, but certainly more opportunities exist.

**Thank you for the positive evaluation of our manuscript. We will take the suggestions into consideration and believe they can be easily incorporated.

Major comments:

Line 194-96: Please provide more details on extraction methods. Reference to Warter (2024) is ok for details; but please briefly outline methodology here as well (were kits used? Which ones? Any QA/QC?)

** We added some additional information to the methodology (L230-239).

Briefly, DNA was extracted using a DNeasy PowerSoil Pro Kit (Qiagen, Hilden, Germany) following the manufacturer's protocol (Handbook 03/2021, HB-2495-005) with the following exceptions: In step 1, 750 µl of CD1 buffer was used, and in step 16, CD6 buffer was replaced with AE buffer from the DNeasy Blood & Tissue kit (Qiagen). AE contains EDTA in order to provide a better conservation of the samples.

Lines 184+: Throughout the Methods section (and/or Results), consider adding a bit more context to the parameters you compare (e.g., briefly, what is the importance/relevance of EC, DIC, etc).

** EC in particular gives an indication of levels of dissolved charged chemicals (i.e. salts, inorganic compounds), which can be linked to the influx of specific water or pollution sources. DIC – as the sum of inorganic carbon species in water, plays a major role in many biological and environmental contexts, in particular in relation to water quality, ecosystem health and carbon cycling.

We added some additional explanations on hydrochemistry (L 210-217).

Line 203+: Details of macrophyte sampling design not available (referenced paper is in review). Please detail here (i.e, what is "qualitative and quantitative inventory")

** The referenced paper has now been published and the detailed sampling design can now be referenced.

Qualitative and quantitative inventory refers to recording of all species present in the water body as well as the estimation of their abundance, respectively.

We made changes to the text to make this clearer (L248-254).

Line 213: It may be worthwhile to describe what is meant by "relative read-abundance data."

**We changed this to "relative abundance data" (L270).

Lines 213 - 225: Rather than just offering a list of statistical tests, consider revising this section to include a bit more narrative and explanation about what you are trying to achieve with each of these statistical tests and why each was selected (over others).

**As per suggestion of reviewer 1, we will add some references to show the applications of some of the statistical test in the context of assessing diversity.

We revised the text to a more narrative structure. We also provided additional references for statistical tests (L259-279).

Line 230: Please define lc-excess (here or in Methods section), possibly with some further explanation (one sentence) about what we can learn from this metric.

**We provided a definition of lc-excess and added additional explanations (L 222-226).

Line 234: Might be worth describing what is meant by "young water," here or in the Methods.

**We added some explanation regarding the definition of young water (L218-222).

Line 246: Table 1 title - Probably appropriate to cite your model here

**We added a reference to the model in the Table title.

Line 290: This sentence is confusing to me (and again in line 293). How does a detected ASV not appear in at least one sample?

**As per suggestion of reviewer 1 added percentages here for easier readability.

We acknowledge an error in the wording and corrected this part. As stated in the methods, only species present in >10% of the samples were retained, which removes a certain number of ASVs – thus resulting in the reduced percentage of total ASVs used for the analysis.

We made some changes to the text to match (L349-354).   what is written in the methods (L243-244).

Line 294: Could you explain how scaling and stress translates to acceptable representation?

**Stress usually refers to the goodness of fit of the NMDS ordination. In general, for a good representation the stress value should ideally be less than 0.2.

Scaling refers to the number of dimensions.

We removed the description of scaling in the plot and in the text to avoid confusion (L354).

Technical/minor corrections:

Many misplaced or awkward commas. Also, some awkward wording and sentence structure. Please carefully edit for punctuation and grammar throughout.

**We acknowledge that some sentences are too long and complex. We checked all punctuation, commas and sentence structure and broke up sentences where needed

Data and Methods:

Line 108: Missing a period after first sentence.

**Changed

Line 109: "Berlin's"

**Changed

Line 112: Consider defining "mbgl" the first time you use this abbreviation.

**We added definitions.

Line 124: Awkward/incorrect punctuation

**Changed.

Line 143: m3/s?

**Changed

Line 148: Unnecessary period

**Changed.

Results:

Line 228: Reference to Figure 2a, but there is only Figure 2.

**Changed

Line 229: Grammatical error with "evaporative"

**We added a missing word – "fractionation". (L284)

Line 272: Define DIC

**We defined this now in the Method section (Section 2.3 – Stable water isotopes and hydrochemistry) (L205).

Line 285: Define ANC

**We defined this now also in the Method section (L214-215).

Line 312: Consider switching the clauses in this sentence so that it reads more smoothly.

**Changed (L372).

Discussion and Conclusion

Concise and generally well-written

**Thank you for this positive feedback.

Figure 1.n Sampling sites are difficult to distinguish on Figure 1c; consider contrasting colors.

**We changed the color coding to black in this figure for better visibility.

Add sampling sites to Figure 1d?

**We added sampling sites in red to Figure 1d.

More info in caption? Sampling for what? Context (city, country, year)?

**We added some more information in the caption (L170-175).

I do not see "urban degraded streams" on the figure. Maybe use consistent language (effluent)?

** Changed

Figure 2. Define lc-excess. More info in caption (location, year?)

**The definition of lc-excess was initially moved to the supplementary material. We now added the definition and background in the method section (L222-226).

Figure 3. Move legends together (type and season) and shared between plots

**Changed

Figure 4. How did you define "outliers" (i.e., what is the threshold?)

** Outliers were defined as values that lie outside the overall distribution pattern, falling outside the ranges indicated in the boxplot. In most cases, these are single samples were certain species exhibited an increased abundance, which was not present in later samples, thus resulting in singular high/low values of alpha diversity measures. No specific threshold was chosen.

Remove period "Coloring corresponds to sample subgroups" Missing an asterisk for significance level (0.01)

**Changed.

Consider simplifying the plot by eliminating p-values for non-significant relationships (particularly if they are reported in the text) Overarching p-values (?) in upper left corner of plots a), b) was confusing to me; could you annotate this something like "All groups p = 0.004.

**We removed insignificant p-values from the main plot. We added clarification to p-values that apply to all groups.

Also, consider putting this in the same place on each plot (e.g., next to the plot letter?)

**Changed.

Figure 5. Typo first sentence: "diatoms/algae" I don't see asterisks for sig level in the plots. If I missed them, perhaps consider a different color (e.g., black?)

**We amended the figure caption. The statement regarding significance was erroneous and has been removed.

Figure 6. Put definition first followed by acronym (DBLM) (also in Fig. 7) Consider using the same colors for seasons as in Figure 1. Is colorramp necessary?

**Yes, per suggestion of reviewer 1, we changed the color coding and symbols to match Figure 1. Abbreviation has been changed.

References:

Space or indent entries to be more readable/distinguishable

**This will be changed during the typesetting process anyway to match the format of the journal.

Supplementary Material

I think developing more descriptive titles and captions for all of these (including defining abbreviations) could be worthwhile.

**We amended captions of tables and figures to be more descriptive.

Table S1. Review and possibly revise caption: mean water level in cm (WL) -> Mean pond depth (m)?

**Changed

Table S6. Move significance level to separate column -OR- re-order parameters by decreasing significance?

**We sorted significance levels in the table from high to low.

Sampling Methods: Consider moving these into the main manuscript text.

**In an effort to reduce the manuscript length, we initially moved these to the supplementary materials. We now moved some essential information such as definition of lc-excess, definition of young water fractions, and definitions of certain chemical parameters back to the main manuscript for better understanding (L 205-226).

---

## Author Response (AR2)

Dear Editor,

we provided additional information as requested pertaining to the young water fraction (L 2016-210). We hope this is satisfactory and gives sufficient context for the reader. We also adapted the color scheme in Fig. 7 to a scheme more fitting for readers with color vision deficiencies.

Thank you for your thorough review!

Kind regards,

Dr. Maria Magdalena Warter (on behalf of all co-authors)